# Real-Life Outcomes of a Multicomponent Exercise Intervention in Community-Dwelling Frail Older Adults and Its Association with Nutritional-Related Factors

**DOI:** 10.3390/nu14235147

**Published:** 2022-12-03

**Authors:** Fernando Millan-Domingo, Francisco Jose Tarazona-Santabalbina, Aitor Carretero, Gloria Olaso-Gonzalez, Jose Viña, Maria Carmen Gomez-Cabrera

**Affiliations:** 1Freshage Research Group, Department of Physiology, Faculty of Medicine, University of Valencia, CIBERFES, Fundación Investigación Hospital Clínico Universitario/INCLIVA, 46010 Valencia, Spain; 2Programa Mejora S.L., 46002 Valencia, Spain; 3Department of Geriatric Medicine, Hospital Universitario de la Ribera, 46600 Valencia, Spain; 4Medical School, Universidad Católica de Valencia San Vicente Mártir, 46001 Valencia, Spain

**Keywords:** reproducibility, Real-Life, frailty, clinical trial, adherence, health-span

## Abstract

Most of the studies on physical exercise in older adults have been conducted through randomized clinical trials performed under tight experimental conditions. Data regarding Real-Life physical exercise intervention programs in older adults with different conditions and in different settings, are lacking. This is an interventional, prospective and pragmatic Real-Life study in which fifty sedentary and frail individuals were enrolled. We aimed at determining if a Real-Life exercise intervention outweighs previously reported improvements in a Clinical Trial (NCT02331459). We found higher improvements in the Real-Life exercise intervention vs. the Clinical Trial in functional parameters, such as Fried’s frailty criteria, Tinetti, Barthel and Lawton & Brody scales. Similar results were found in the dietary habits, emotional and social networking outcomes determined through the Short-MNA, Yesavage, EuroQol and Duke scales. The Real-Life intervention led to a significant reduction in the number of falls, visits to the primary care centers and emergency visits when compared to the results of our previously published Clinical Trial. The implementation of a Real-Life exercise intervention is feasible and should be a major priority to improve health-span in the older population.

## 1. Introduction

Application of clinical trial results to clinical practice is not straightforward [1]. Issues such as reproducibility and replicability due to restrictive enrolment criteria, experimental design limitations, financial issues and biological variability can all underlie the disparity between the outcomes achieved in clinical trials compared to those achieved in clinical practice.

Replicability and reproducibility are fundamental assumptions in science. Clinicians are generally interested in the replicability of a trial that refers to the ability to obtain consistent results across studies aimed at answering the same scientific question, each of which has obtained its own data [2]. According to Popper, “Non-reproducible single occurrences are of no significance to science” [3]. From the perspective of a practicing physician, reproducing the precise experimental conditions of a trial in ordinary clinical setting is, in most cases, very difficult. Clinical trials are characterized by strict control of the variables under study. This is far from being achieved in Real-Life clinical practice, where the heterogeneity of patients and experimental settings makes this control challenging.

There is evidence from several research groups, including our own, showing that physical exercise is an effective treatment for functional improvement in older adults [4,5,6,7,8,9,10]. A multicomponent exercise program, defined as a combination of strength, coordination, neuromotor, cardiorespiratory and flexibility exercises, is the best choice to enhance functional parameters in the geriatric context [10,11,12,13].

In addition, inadequate nutrition is an important determinant of frailty, disability and death [14,15,16].

Up to now, most of the published studies on physical exercise in older adults have been developed through randomized clinical trials or have been performed under tight experimental conditions. Data from Real-Life interventions showing results of physical exercise programs among older adults with different conditions and in different settings, are lacking [4]. Real-life studies have been defined as interventions in everyday settings that provide information on the Real-Life context [17]. Within the different types of Real-Life studies, a pragmatic trial one that conducts a methodology in routine locations which increase the likelihood of obtaining relevant conclusions to clinical practice.

In this study we aimed at comparing the results from a successful clinical trial (NCT02331459) previously published by our research group [10] with a Real-Life intervention in four different locations of the Valencian area, in Spain. The geographical proximity between these locations prevents major differences in the eating habits and routines of the elderly recruited in the study. The clinical trial was developed through primary health centers by the Department of Research in the health department of the University Hospital of La Ribera, whereas the Real-Life intervention took place outside the hospital setting and independently of the primary care centers. The same inclusion criteria were used in both studies.

Our aim was to compare the results obtained, in terms of frailty, functionality, cognitive, emotional and nutritional status and social networking in a strictly controlled randomized clinical trial with those of a similar Real-Life intervention where such a tight control could not be performed.

## 2. Materials and Methods

### 2.1. Study Design and Participants

This is an interventional, prospective and pragmatic Real-Life study that has been compared with a previously published interventional, controlled, randomized clinical trial (NCT02331459) [10].

A sample size of 45 participants was calculated considering a difference of 6 points in the Barthel index between the groups, with a variance of 100. A difference of 5 points on the Barthel Index between the groups was used in our previous RCT [10]. We increased the sample size to 50 patients that were recruited from four different locations. We contacted the city councils of four towns in the Valencia area: Picanya, Alaquas, Cofrentes and the Trinidad neighborhood (located in Valencia city). Through the social services, the town hall staff contacted potential participants by telephone and then invited them for an interview to provide more details and to check the eligibility criteria. The 50 participants recruited were compared with the 100 patients that took part in a previous clinical trial (NCT02331459), where the sample was divided into two groups: control and intervention [10].

Sample characteristics at study entry are provided in Appendix A
Table A1. The town’s own sports facilities were used for the evaluations and the development of the intervention program.

The protocol was approved by the Committee on Ethics in Research of the Faculty of Medicine, University of Valencia (Reference number: H1500375577317).

This study took place between September 2018 and February 2020. Informed consent was obtained from each participant, who signed after fully understanding the procedures.

The eligibility criteria were the same as those used in the previous Clinical Trial [10]: men and women aged 70 years or older who were (1) sedentary, (2) frail according to the frailty phenotype, (3) with a gait speed slower than 0.8 m/s and (4) community dwellers. Exclusion criteria were (1) life expectancy of less than 6 months by any cause, (2) cognitive impaired patients (score < 17 in the Mini-Mental State Examination [MMSE]), (3) severe disability (score < 40 points on the Barthel Scale), (4) hospital admission in the past 3 months for any reason, (5) oncologic patients with active chemotherapy or radiotherapy treatment, (6) major non-ambulatory surgery in the past 6 months before the beginning of the study, (7) coronary events in the past 12 months, (8) institutionalized patients, or (9) impossibility to go to the sport center when using their own means of transport.

A diagram outlining subject flow is provided in Figure 1.

### 2.2. Multicomponent Exercise Program

In our previous Clinical Trial [10], the multicomponent exercise intervention lasted 65 min 5 days per week for 24 weeks. Briefly, the sessions were delivered in groups, were supervised and involved a combination of the following activities: proprioception and neuromotor exercises (10–15 min), cardiorespiratory training (initially at 40% of maximum heart rate increasing progressively to 65%), strength training (initially at 25% of 1 repetition maximum up to 75%) and stretching. Patient exercise compliance was 47.3% (95% confidence interval [CI] 38.7%–55.7%). The neuromotor exercises included postural sway and dynamic balance, coordination and flexibility of the lumbo-pelvic area. The cardiorespiratory training included walking around a circuit and climbing stairs. The strength training was performed with resistance bands and included isometric, concentric and eccentric exercises with arms, hands and legs. The stretching exercises included arms, legs and neck. The details of time, intensities and progression of the exercise training can be seen in [10]. The ratio of trainers to participants was 1:15.

In the Real-Life study intervention, the sessions were conducted and supervised by a sport scientist, in group and included five minutes of warming-up, 20 min of strength exercises from week 1 to week 8 at 45%–55% of one-repetition maximum, from week 9 to week 16 at 65% one-repetition maximum, from week 17 to week 24 at 70%–75% one-repetition maximum; 20 min of cardiorespiratory exercises from week 1 to week 8 at 55% HRmax, from week 9 to week 16 at 65%–70% HRmax, from week 17 to week 24 at 70%–75% HRmax; 10 min of neuromotor training and 5 min of stretching (Table A3 and Table A4).

It has been shown that exercise programs that seem to result in better outcomes are those performed 3 days per week [18]. Thus, the Real-Life intervention was performed 3 days per week, for 60 min, for 24 weeks.

Heart rate was monitored and supervised in every participant by the sport scientist during all the training programs. A heart rate higher than that designed for the aerobic exercise, dizziness symptoms and muscle or joint pain were criteria to stop the intervention.

Briefly, the main changes made in the Real-Life intervention lie in the number of weekly sessions and the fact that the participants train all the physical capacities in each session. The type of exercise, the materials used and the trainer to participant ratio were the same as in our previous RCT (1:15).

### 2.3. Measurements

The following parameters were recorded: age, gender, social situation, marital status. Anthropometric data: abdominal, brachial and leg girths with a SECA anthropometric belt; lean mass; and fat mass percentages were determined by bioelectrical impedance analysis (Tanita. Inner Scan V BC-601). Functional assessment included: Barthel Index (basic activities of daily living), Lawton & Brody (instrumental activities of daily living), Tinetti (fall risk) and hand grip strength with a Jamar (c) Hydraulic Hand Dynamometer. Cognitive, emotional and social determinations were assessed using the MMSE, Duke social support, EuroQol quality-of-life scale (EQ-5D) and geriatric depression scale from Yesavage. We also determined frailty [19,20] and the nutritional status of the individuals with the Short-MNA scale [21].

Prevalence of other geriatric syndromes, number and risk of falls, number of voluntary hospital admissions, visits to the emergency service and visits to the primary care center in the previous 6 months were also recorded.

Clinical measurements included comorbidities, pulsi-oximetry, resting EKG, arterial hypertension, renal insufficiency, fear of falling (Falls Efficacy Scale, FES).

All the data were registered using an Apple iPAD, stored at Microsoft’s Azure cloud and protected with a VPN.

### 2.4. Statistical Analysis

Statistical analysis was performed using the GraphPad Prism 8 software. Categorical variables were described as the frequency and percentage, and quantitative variables as the mean and standard deviation (SD). Descriptive analyses were carried out for each of the two groups. The between-group differences in the frequency distribution across categories were analyzed using the χ2-test, whereas the mean differences between groups were analyzed using the independent-samples *t*-test with quantitative variables that showed a normal distribution with the Shapiro-Wilk test, while the variables with non-normal distribution were treated with the non-parametric Wilcoxon test for paired data. An ANCOVA analysis was also performed for the main outcome variables using exercise time and adherence to the intervention program as confounding variables.

The threshold for statistical significance was established at a bilateral α value of 0.05.

## 3. Results

### 3.1. Effect of a Supervised, Personalized and Social Exercise Program on Age-Associated Frailty

Figure 2a shows that frailty decreases significantly in the Real-Life exercise intervention. These results are even more pronounced than those achieved in the Clinical Trial intervention group [10]. No significant differences in the basal state of frailty were found between groups before the intervention. However, after the intervention, the Real-Life group shows higher improvements when comparted to the Clinical Trial and the Control Group (see Table 1). The clinical characteristics of the different groups of patients are shown in Table A1. This shows that most of the baseline characteristics of the patients were not altered in the different groups studied. Even if we randomly selected patients with the same characteristics, we observed that the number of falls previous to the Real-Life intervention happened to be higher than in the Clinical Trial, as was the hyperlipidemia. On the other hand, the control group seemed to have a statistically significant lower number of smokers. This is obviously a limitation of the study, but we believe that its biological significance does not hinder the validity of the results and the conclusions achieved in our study.

### 3.2. A Real-Life Exercise Intervention Improves Basic as Well as Instrumental Activities of Daily Living

Maintaining both basic and instrumental activities of daily living in their patients is a major undertaking for geriatricians. Basic and instrumental activities of daily living were determined using the Barthel and Lawton & Brody Scales, respectively. Both scales were improved after the Real-Life or the Clinical Trial exercise interventions when compared with the participants’ basal values (see Figure 2b,c). The improvements were more pronounced in individuals who performed the Real-Life intervention than in those involved in the Clinical Trial. Individuals who did not perform exercise got worse on these scales. Table 1 also indicates that, after the 6-month intervention, the Real-Life and Clinical Trial groups show significant improvements when compared to the Control Group in terms of basic and instrumental activities of daily living.

### 3.3. Enrolling in an Exercise Program Improves Nutritional Habits in Older Adults

The Short-MNA scale determines the risk of malnutrition. This is a useful tool to understand the dietary habits of a population whose nutrition may be inadequate for a variety of reasons.

Figure 3 shows that the participants in the Real-Life and Clinical Trial studies scored higher on the Short-MNA scale after the interventions. Subjects in the control group showed lower values in the mini nutritional assessment scale.

### 3.4. Exercise Lowers the Number of Falls and Mitigates the Fear of Falling in a Real-Life Intervention

Figure 4a shows that an exercise program lowers the risk of falls when the gait and balance (Tinetti scale) data is compared between the basal values and those obtained after a 6-month intervention (pre vs. post). Similarly, the participants who did not perform exercise had an increased risk of falling. Table 1 also indicates that, after the 6-month intervention, the Real-Life and Clinical Trial groups show significant improvements when compared to the Control Group.

We determined the number of falls in our populations before and after the interventions. Individuals in the control group showed an increase in the number of falls in the 6 months period studied. On the contrary, the participants involved in the Clinical Trial and those who were engaged in the Real-Life intervention showed a very significant decrease in the number of falls (see Figure 4b). The effectiveness of the intervention in reducing the number of falls was again more pronounced in the case of the Real-Life than in the Clinical Trial intervention.

An important psychological factor to be considered is the fear of falling. Falls are such an intense threat to the health and wellbeing of the old population that the fear of falling is a significant cause of concern. Persons who did not exercise (control) did not experience a lowering in their fear of falling (see Figure 4c). In contrast, those who exercised did lower it both in the Clinical Trial and the Real-Life intervention. The lowering of the fear of falling was significantly more pronounced in the Real-Life than in the Clinical Trial intervention groups.

### 3.5. Real-Life Exercise Intervention Improves the Quality of Life in Old Adults

Exercise resulted in a clear improvement in quality of life (as determined by the EQ-5D scale). Figure 5a shows that patients who did not perform exercise significantly lost perceived quality of life in the six months trial duration. On the contrary those who performed the exercise significantly increased their quality of life.

The Duke scale is a questionnaire for the social support perceived by the patient. Again, those who did not follow the exercise intervention significantly lost social support. No changes in social support perception were found in the participants from the Clinical Trial exercise group, whereas those who exercised in the Real-Life intervention improved their social support (see Figure 5b).

Figure 5c shows that participants from the Clinical Trial control group significantly increased their depressive state as determined by the Yesavage scale. The exercise intervention both in the Clinical Trial and in the Real-Life studies very significantly improved the participant’s depressive state. After the intervention, the comparison between groups shows how the reduction of depression criteria, as well as the increase in the perception of quality of life, were more evident in the Real-Life group. However, in perceived social support no significant differences were found (see Table 1).

### 3.6. Real-Life Exercise Intervention Can Result in Substantial Savings in Healthcare Cost Expenses

Reductions in the number of emergency visits or in visits to primary care centres does not only reflects improvements in the health status of frail individuals but also results in a significant reduction in public health expenditure. As seen in Figure 6a, the number of visits to primary care centres decreased in the Clinical Trial intervention group. Moreover, this number was further reduced in Real-Life intervention participants. The reduction in the visits to primary care centres of the Real-Life group resulted in a significant difference when compare with the Control Group after the intervention (see Table 1).

Equally important is the number of visits to emergency care centres. These were not decreased in the clinical trial exercise intervention group but were significantly decreased in the Real-Life exercise intervention participants (Figure 6b). The Real-Life group also shows a significant decrease in the number of visits to emergency care centres after the intervention when compared with the Control Group (Table 1).

The data shown in Figure 6 point towards the critical importance of exercise in lowering public health expenditure for the ever-growing numbers of frail old persons in our society.

The exercise intervention also improved grip strength and anthropometric parameters such as lean mass and fat mass percentages and abdominal and brachial girths (See Figure A1, Figure A2 and Figure A3).

### 3.7. Effect of Time and Adherence to the Intervention in the Main Outcomes of the Real-Life Intervention

Using time and adherence as confounding variables and making a global analysis, we can say that the outcomes obtained with the Real-Life intervention outweigh those obtained in the Clinical Trial (see Table 1).

The Real-Life intervention reduced Fried’s frailty criteria to a greater extent than the clinical trial groups. We also found a more pronounced increase in both dominant and non-dominant hand grip strength.

Due to the intra-group variability inherent in the Real-Life interventions, we found smaller differences in the post-intervention comparison between groups for the Barthel and Lawton & Brody functional scales. However, the improvements due to the Real-Life intervention in the instrumental activities of daily living are even greater than those achieved in the Clinical Trial group.

## 4. Discussion

### 4.1. Short-Term Effects of a Multicomponent, Social, Personalized and Supervised Exercise Program

Life expectancy has been increasing at approximately 2 years per decade for the last 150 years [22]. Much of this increase, especially from the second half of the 20th century, has been due to increased survival of middle-aged people into old age, but not all of this life extension is spent in good health. An average woman or man with a life expectancy of around 82 years can expect to live 19 of those years (~20%) in poor health [23]. Frailty is a good valuable target in attempts to improve the health-span [19].

There is growing evidence that frailty is not an inevitable and unalterable process; on the contrary, it is amenable to intervention [24,25,26,27].

Any intervention to delay the onset of frailty and, most importantly, the transition from frailty to disability, should of course be effective, but it is important that the beneficial effects are seen in the relatively short term.

It is very well established that life-long practice of salutary habits results in a prolongation of life- and health-span [25,26], but in the clinic, we frequently find persons who are above the age of 70 and who have not carried out a life of salutary habits, especially physical activity. A major question is whether a short-term exercise program could significantly improve the health=span of these persons. In our previous reports on a Clinical Trial [10], we showed that relatively short-term (six months) exercise training results in a significant improvement in health-span in old frailty individuals.

Our results on a Real-Life study show that, with a personalized, multicomponent, supervised and social intervention, we can outweigh the improvements achieved in a Clinical Trial in terms of quality of life of individuals at risk of becoming disabled. Moreover, we have found that enrolment in an exercise program improves nutritional habits in older adults. The participants in the Real-Life and Clinical Trial exercise studies scored higher on the Short-MNA scale after both interventions. It is well known that physical exercise effects on fitness are influenced by nutritional status. The cross-talk between these two lifestyle factors, exercise and nutrition, during aging deserve further research and attention [28].

The age-associated loss of function is intrinsic to all the cellular systems. However, the decline in muscle mass and function, preferentially of our lower body, probably represents the most dramatic and significant of all changes during the aging process [29,30,31]. Muscle power begins to decline after the age of 30 and continues to decline linearly with advancing age [32]. From the age of 50, there is a progressive loss of muscle mass (1–2% per year) and of muscle strength (2–5% per year) [33,34] with clinical consequences. Frailty and other age-related diseases increase muscle catabolism which has important implications in metabolic diseases [35]. The results from our Real-Life intervention showed that there was an improvement in body composition, an increase in muscle mass and a decrease in fat mass (See Table 1 and Figure A2 and Figure A3). More importantly, it was accompanied with improvements in grip strength (See Table 1 and Figure A1). These changes have relevant clinical implications, muscle strength is a strong predictor of slow gait speed, severe mobility limitation, risk of falls and hospitalization and high mortality rate [36]. Older adults with low muscle strength have ~2-fold greater risk of mortality compared to stronger old individuals [37].

### 4.2. A Real-Life Exercise Intervention Improves Adherence to the Exercise Program

The optimal frequency for a multicomponent exercise program aimed at the frail older adult is 2–3 days a week [9,38]. It is beneficial for all outcomes, but the physical and psychosocial determinants show the bigger improvements [18]. We have found that a Clinical Trial exercise intervention significantly improves functional parameters in old frail individuals. Interestingly, an adaptation of this strictly controlled randomized clinical trial to a Real-Life setting results in even better outcomes in terms of functionality for the participants.

One of the main adaptations of the Real-Life exercise program was the reduction from five to three exercise sessions a week, with an adherence of 79% (95% confidence interval [CI] 72%–86%), while the adherence to the Clinical Trial program was 47% (95% confidence interval [CI] 39%–56%).

This modification means that, on average, a participant in the Real-Life exercise intervention attended a total of 57 sessions out of 72, while the patients in the Clinical Trial attended an average of 56 sessions out of a total of 120.

The main characteristic of a Real-Life intervention in physical exercise is that the development of the sessions is performed in an everyday environment and close to the participant’s home. This accessibility of the program leads to a greater adherence. In agreement with the results found in our study, other pragmatic Real-Life interventions have also reported higher adherence when compared to clinical trials carried out in groups with similar characteristics [4,39].

We also think that the superior benefits achieved with the Real-Life intervention could be explained by the fact that all the physical capacities were trained in every session by the participants in this program, while in the Clinical Trial a different one was trained each day. This allowed a better adaptation to training in the Real-Life intervention [40].

These results, together with those of the recently mentioned pragmatic interventions [4,39], confirm that the implementation of studies in everyday locations is favorable for participants. Moreover, they increase adherence to the interventions, establishing them as an important focus for further research in the Real-Life context.

### 4.3. Economic Impact of the Real-Life Exercise Intervention

The cost of caring for a disabled person is at least 16 times more than the cost of caring for a vigorous one [41].

It is obvious that a major aim for health sciences, including medicine, nursing, physical activity, etc., is to prevent the transition from vigorousness to frailty and from frailty to disability. The figures become especially impressive when one thinks in terms of one given country. For instance, in Spain, in August 2020, there were over 1,346,000 individuals who were disabled out of a total population of approximately 47 million citizens. If we consider that the cost or caring for each one of these disabled persons is, as previously discussed, 14,000€ per year, then the cost of disability in a country like Spain amounts to 18 billion euros per year. In accordance with the health cost rates in the Valencian region the Real-Life exercise intervention resulted in a reduction in spending of 16,628€. This is more than the total reduction in health costs of the clinical trial intervention (11,163€). Table A2 shows that this saving is driven by both reduction in the number of visits to primary care centres and to emergency wards. The reduction in health expenses due to the 6-month Real-Life or the Clinical Trial intervention were −16,629€ (−56.2%) and −11,163€ (−38.7%), respectively [42]. However, those patients that did not follow the exercise program ended with an increase in the average cost of the primary care and emergency visits from 21,485€ to 25,676€ in just half a year [42].

Any measure that we may take, like the ones we describe in this paper, to lower the cost of disability will mean not only a tremendous improvement in human well-being, but also a very serious saving in social costs.

## 5. Conclusions

Adherence to a Real-life, social, personalized and supervised multicomponent exercise program results in remarkable improvements in terms of well-being, nutritional habits and in the reduction in health costs. The practical application of the multicomponent physical exercise program resulted in better results than those previously obtained in a randomized Clinical Trial. The implementation of this type of intervention should be a major priority for social security and health services.

## Figures and Tables

**Figure 1 nutrients-14-05147-f001:**
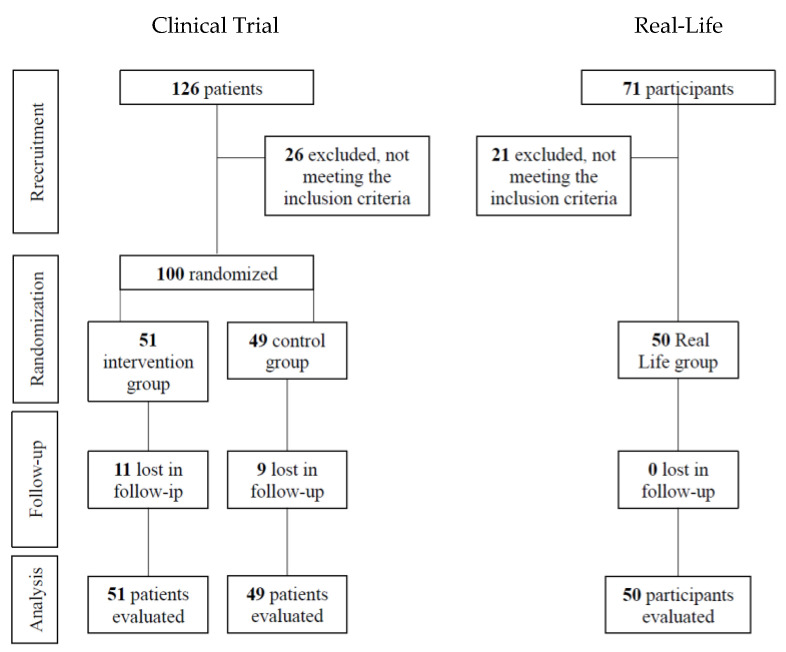
Flow diagram of the participants.

**Figure 2 nutrients-14-05147-f002:**
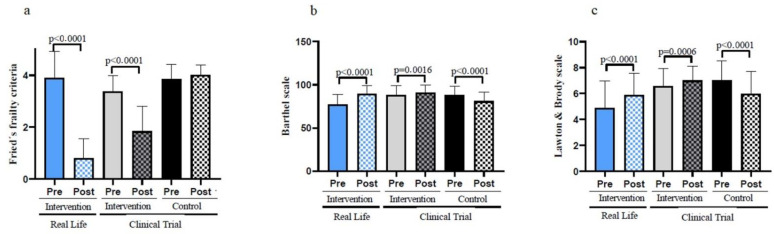
Functional parameters in old individuals before and after a six-months intervention. (**a**) Number of frailty criteria fulfilled by the participants before and after the Real-Life and the Clinical Trial interventions. (**b**) Performance in the basic activities of daily living using the Barthel Scale. (**c**) Performance in the instrumental activities of daily living using the Lawton & Brody Scale. Real-Life Intervention *n* = 50, Clinical Trial Intervention *n* = 51, Clinical Trial Control *n* = 49. Bars represent mean ± SD. Statistical analysis was performed with the Student’s *t* test for paired samples and the Wilcoxon test for non-parametric samples.

**Figure 3 nutrients-14-05147-f003:**
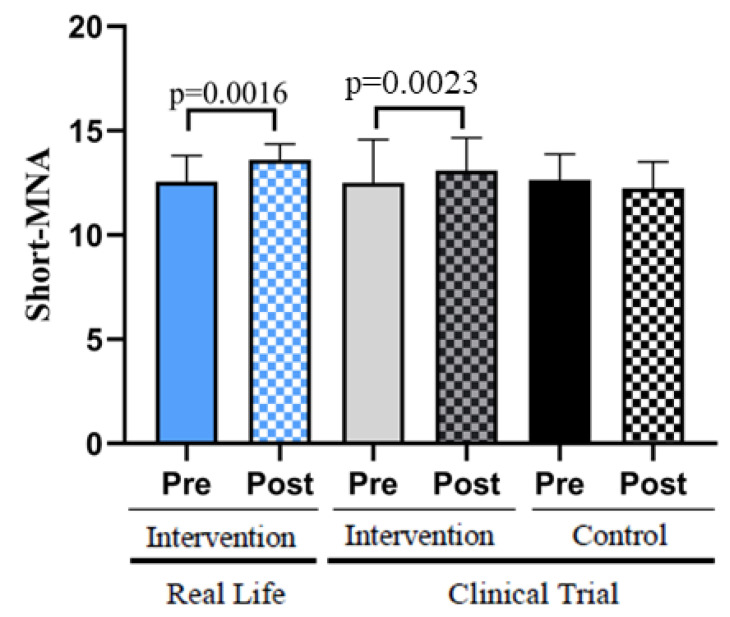
Nutritional status assessed with the Short-MNA Scale. Real-Life Intervention *n* = 50, Clinical Trial Intervention *n* = 51, Clinical Trial Control *n* = 49. Bars represent mean ± SD. Statistical analysis was performed with the Student’s *t* test for paired samples and the Wilcoxon test for non-parametric samples.

**Figure 4 nutrients-14-05147-f004:**
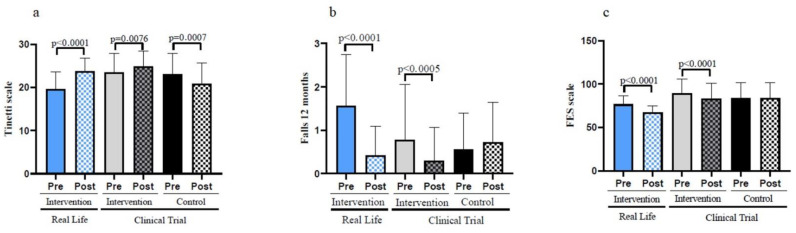
Fall risk, number of falls and fear of falling assessed in old individuals before and after a six-month intervention. (**a**) Assessment of the Tinetti scale. (**b**) Number of falls before and after the Real-Life and the Clinical Trial interventions. (**c**) Fear of falling before and after the Real-Life and the Clinical Trial intervention. Real-Life Intervention *n* = 50, Clinical Trial Intervention *n* = 51, Clinical Trial Control *n* = 49. Bars represent mean ± SD. Statistical analysis was performed with the Student’s *t* test for paired samples and the Wilcoxon test for non-parametric samples.

**Figure 5 nutrients-14-05147-f005:**
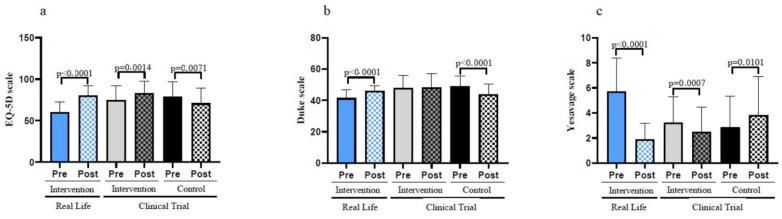
Self-rated emotional and social assessments before and after the interventions in old individuals. (**a**) Evaluation of the quality-of-life by using the EuroQol scale. (**b**) Results on the Duke Scale questionnaire for the social support perceived by the patient. (**c**) Results on the geriatric depression scale of Yesavage. Real-Life Intervention *n* = 50, Clinical Trial Intervention *n* = 51, Clinical Trial Control *n* = 49. Bars represent mean ± SD. Statistical analysis was performed with the Student’s *t* test for paired samples and the Wilcoxon test for non-parametric samples.

**Figure 6 nutrients-14-05147-f006:**
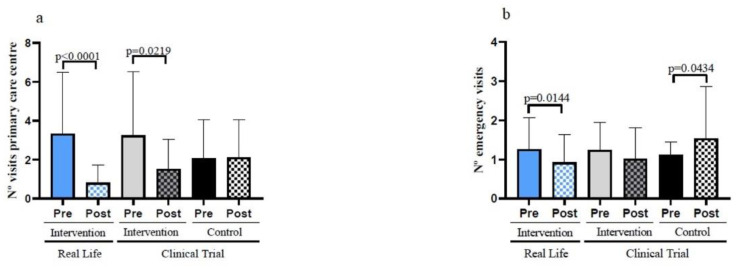
Hospital visits in the previous and later six months of the interventions in the study participants. (**a**) Number of visits to the primary care centres before and after the interventions. (**b**) Number of emergency visits before and after the interventions. Real-Life Intervention *n* = 50, Clinical Trial Intervention *n* = 51, Clinical Trial Control *n* = 49. Bars represent mean ± SD. Statistical analysis was performed with the Student’s *t* test for paired samples and the Wilcoxon test for non-parametric samples.

**Table 1 nutrients-14-05147-t001:** Changes in the main outcomes analyzed in our study before and after the interventions, Real-life and CT.

	Real-Life Intervention (Pre) *n* = 50	Clinical Trial Intervention (Pre) *n* = 51	Control (Pre) *n* = 49	*p* Value	Real-Life Intervention (Post) *n* = 50	Clinical Trial Intervention (Post) *n* = 51	Control (Post) *n* = 49	*p* Value
Frailty	4.0 (SD 1.1)	3.6 (SD 0.8)	3.8 (SD 0.6)	^a^ 0.360 ^b^ 0.470 ^c^ 0.890	0.8 (SD 0.4)	1.8 (SD 1.0)	4 (SD 0.8)	^a^ <0.001 ^b^ <0.001 ^c^ <0.001
Lean mass%	51.0 (SD 51.1)	44.2 (SD 8.3)	48.4 (SD 9.8)	<0.001 0.321 0.027	53.1 (SD 4.8)	44.5 (SD 8.8)	44.3 (SD 10.9)	<0.001 <0.001 1.000
Fat mas%	33.5 (SD 7.1)	35.9 (SD 10)	37.2 (SD 10.3)	0.552 0.088 1.000	29.1 (SD 4.3)	30.8 (SD 8.8)	37.8 (SD 8.4)	0.791 <0.001 <0.001
BRA girth (cm)	31.6 (SD 3.6)	30.1 (SD 3.8)	30.0 (SD 2.9)	0.321 0.180 1.000	30.6 (SD 2.8)	29.2 (SD 3.8)	29.7 (SD 3.2)	0.127 0.694 1.000
ABD girth (cm)	90.4 (SD 18.0)	104.0 (SD 15.9)	105.5 (SD 9.3)	<0.001 <0.001 1.000	87.1 (SD 15.2)	100.6 (SD 15.9)	104.8 (SD 16.3)	<0.001 <0.001 0.052
Weight (kg)	70.9 (SD 12.2)	74.2 (SD 13.3)	74.6 (SD 13.2)	0.506 0.524 1.000	69.5 (SD 11.7)	73.2 (SD 12.7)	73.4 (SD 12.2)	0.373 0.328 1.000
Dominant hand Grip (kg)	19.3 (SD 4.5)	20.2 (SD 8.6)	20.5 (SD 4.7)	1.000 1.000 1.000	22.5 (SD 4.1)	21.2 (SD 8.6)	18.4 (SD 5.2)	0.845 0.004 0.891
Non-dominant hand grip (kg)	18.6 (SD 4.2)	20.1 (SD 8.1)	18.8 (SD 5.3)	0.548 1.000 0.650	21.4 (SD 4.3)	20.6 (SD 7.4)	17.9 (SD 5.3)	1.000 0.010 0.054
Tinetti scale	19.6 (SD 4.1)	23.5 (SD 4.4)	23.1 (SD 4.8)	0.056 0.069 1.000	23.7 (SD 3.1)	24.7 (SD 3.6)	20.9 (SD 4.8)	0.861 0.002 <0.001
Nº Falls	1.6 (SD 1.8)	0.8 (SD 1.3)	0.6 (SD 0.8)	0.003 <0.001 0.666	0.4 (SD 0.7)	0.3 (SD 0.7)	0.7 (SD 0.9)	0.275 1.000 0.078
FES scale	76.5 (SD 10.4)	89.0 (SD 16.9)	83.7 (SD 18.2)	<0.001 0.0632 0.328	67.4 (SD 7.7)	83.2 (SD 12.2)	83.6 (SD 18.5)	<0.001 <0.001 1.000
EQ-5D	59.9 (SD 12.5)	74.5 (SD 17.7)	78.7 (SD 17.8)	<0.001 <0.001 0.416	80.5 (SD 11.2)	82.5 (SD 14.8)	71.3 (SD 18.4)	1.000 0.009 <0.001
EM visits	1.3 (SD 0.8)	1.2 (SD 0.7)	1.1 (SD 0.3)	0.832 0.567 0.732	0.9 (SD 0.7)	1.0 (SD 0.7)	1.5 (SD 1.3)	0.075 <0.001 0.104
PC visits	3.3 (SD 3.1)	3.2 (SD 3.3)	2.0 (SD 1.9)	1.000 0.101 0.191	0.8 (SD 0.9)	1.5 (SD 1.4)	2.1 (SD 1.9)	0.067 <0.001 0.078
Barthel scale	77 (SD 11.9)	87.9 (SD 10.9)	88.2 (SD 10.4)	<0.001 <0.001 1.000	89.7 (SD 9.1)	90.8 (SD 8.5)	81.1 (SD 10.6)	1.000 <0.001 <0.001
Yesavage scale	5.7 (SD 2.6)	3.2 (SD 2.1)	2.8 (SD 2.5)	<0.001 <0.001 1.000	1.9 (SD 1.3)	2.5 (SD 1.9)	3.9 (SD 3.7)	0.561 <0.001 0.005
Duke scale	41.4 (SD 5.2)	47.7 (SD 48.1)	48.9 (SD 6.7)	<0.001 <0.001 0.845	46.1 (SD 3.4)	48.1 (SD 9.1)	44.0 (SD 6.4)	0.350 0.372 0.006
Lawton & Brody scale	4.9 (SD 2.1)	6.6 (SD 1.4)	7.0 (SD 1.5)	<0.001 <0.001 0.740	5.9 (SD 1.6)	7.0 (SD 1.1)	5.9 (SD 1.7)	<0.001 1.000 0.002

^a^ Real-Life-Clinical Trial. ^b^ Real-Life-Control. ^c^ Clinical Trial-Control. Frailty: Linda Fried Frailty Criteria. BRA girth: Brachial girth. ABD girth: Abdominal girth. FES scale: Fear Efficacy Scale. EM visits: Emergency visits. PC visits: Primary Care visits.

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
