# Peer review of "Real-Life Outcomes of a Multicomponent Exercise Intervention in Community-Dwelling Frail Older Adults and Its Association with Nutritional-Related Factors"

_nutrients, 2022, doi:10.3390/nu14235147_

Round 1

Reviewer 1 Report (Previous Reviewer 3)

The results include differences between baseline and follow-up within groups, but this analysis does not appear to be described in the statistical analysis section.  Emphasis in the results should be placed on between groups and not within groups.  

The term "simple" is used to describe the study.  It is uncertain what this means.  Suggestion is to remove this term.  

It is uncertain if authors consulted a statistician as suggested by reviewer 2 (comment #4), as the violation of assumptions of ANOVA were not addressed in the manuscript or response to the reviewer.  

A number of grammatical errors exist making it difficult to read the paper.  

Author Response

We thank reviewer 1 for his/her critical evaluation and valuable comments. We have considered all the recommendations and suggestions. Itemized responses are listed below. All the modifications have been included in the manuscript and marked in red to make the revision easier.

Reviewer 1 comments:

  1. The results include differences between baseline and follow-up within groups, but this analysis does not appear to be described in the statistical analysis section.  Emphasis in the results should be placed on between groups and not within groups.  

Answer 1: We thank the reviewer for this valuable comment. In the new version of the manuscript the differences between groups have been included in the Results section of the manuscript.

  1. The term "simple" is used to describe the study.  It is uncertain what this means.  Suggestion is to remove this term.  

Answer 2: We have removed the term simple.

  1. It is uncertain if authors consulted a statistician as suggested by reviewer 2 (comment #4), as the violation of assumptions of ANOVA were not addressed in the manuscript or response to the reviewer.

Answer 3: We consulted and statistician. An ANCOVA instead of an ANOVA analysis was performed in the new version of the manuscript following reviewer 2 suggestions. We apologize because by mistake we left a sentence in the text including the ANOVA analysis. This mistake has been corrected in the new version of the manuscript

  1. A number of grammatical errors exist making it difficult to read the paper.  

Answer 4:  We have reviewed the entire manuscript.

Reviewer 2 Report (Previous Reviewer 1)

I have understood the author's revisions. 

Author Response

We thank the reviewer for his/her comments 

This manuscript is a resubmission of an earlier submission. The following is a list of the peer review reports and author responses from that submission.

Round 1

Reviewer 1 Report

 The authors have focused that the implementation of a Real-Life exercise intervention in feasible and should be a major priority to improve health-span in the older population. The suggestion in this review is shown as below. 

1.     In the introductions, background information of this study was a little difficult to understand. Are there any different eating or exercise habits from other areas in community-dwelling frail older adults of four different locations of the Valencian area?

2.     The authors showed the change in grip strength, and anthropometric parameters such as lean mass and fat mass percentages and abdominal and brachial girths by the exercise intervention. The information about how their measurements were performed would be better included a little more.

3.     Discussion about how the Real-Life exercise program improved frailty in old individuals from metabolic aspects will strengthen the manuscript.

Reviewer 2 Report

Comments and Suggestions for Authors:

The idea that interventions proposed in research studies may not be applicable in the real-world is an important issue raised by the authors.

 Introduction:

·       Issues raised by the authors such as reproducibility of research findings and their application to clinical settings is relevant to scientists. However, I think the authors place too much emphasis on this issue in the Introduction. The results of their study don’t address reproducibility of scientific findings, but rather demonstrate a different approach to exercise programming in older adults. The authors need to compare and contrast the studies found in the literature that are related to the study they propose.

Methods and Materials:

·       The importance of using statistical significance as a means to determine clinical importance is debatable, but I think sample size estimates are important. However, the authors do not mention where the variance and effects size estimates come from (variance of 100, effects size of 6). What dependent variable? How were those estimates chosen? Was the study powered to detect differences between all three groups?

·       The goal of the study (which is partially relies on secondary data) is to compare two different exercise programs to a control (differences between all three programs). However, it’s very difficult to compare the two exercise programs. The clinical trial (325 min of activity per week over 24 weeks) was quite different from the Real Life (180 min of activity per week for 24 weeks), so it is impossible to tease out whether programs may have been different due to variables such as total energy expenditure. However, the adherence to the programs (47 vs. 79%, respectively) were also different, which makes them more similar. The clinical trial involved (page 3) “proprioception and neuromotor exercises, aerobic, cardiorespiratory training, strength training, and stretching.”, while the Real Life involved (page 4) “strength exercises, cardiorespiratory exercises, neuromotor training and stretching”. Unfortunately, I cannot access the supplementary tables A3 and A4. Can the authors more clearly justify how these programs are similar or different? In other words, if the programs are different from each other in the results, how can they explain those differences?

·       The use of ANOVA for these results is inappropriate. The authors need to utilize a statistical model that integrates the pre-study measurements as independent variables to account for any pre-existing differences between the groups (e.g. ANCOVA). For example, on page 4 the authors state “Figure 2a shows that frailty decreases significantly in the Real-Life exercise intervention. These results are even more pronounced than those achieved in the Clinical Trial intervention group”. However, it’s not stated if there was a treatment x time interaction between the groups. Also, the use of ANOVA does not allow comparisons between the exercise groups or the control group. The percent change tables (2b,d,f; and throughout) are inappropriate and unnecessary if the correct statistical model is used (they need to be removed). I  also have concerns about utilizing ANOVA when there appears to violations of assumptions such as normality and equality of variance (for example, Figure 4c appears problematic). Overall, I highly recommend that the authors consult with a statistician prior to revision for their results and interpretation.

Reviewer 3 Report

This manuscript attempts to examine a real-life physical activity community intervention in older adults on a variety of health related outcomes, including cognition, physical function, falls, mental health, etc.  While the question of the translation of an RCT to a community setting is critical, it seems the question is not whether real-life is better than a RCT, but does a real-life physical activity intervention provide benefits to participants.  With the studies occurring at different times and places, analysis to compare the outcomes seems troublesome, especially considering the differences, albeit potentially minor, in the intervention.  

Other comments include

1.  defining Real-Life.  Use of the term "pragmatic" is more acceptable.  

2. the abstract does not provide any of the analysis comparing the different intervention deliveries

3.  Differential adherence to the programs is a key issue and should be a main topic of discussion and addressed in the introduction as a potential reason to conduct a comparison

4.  the power calculations suggested the need for 50 per group, but 100 participants from the RCT was used in the analysis

5.  it is debatable if real-life exercise interventions include supervisions and in group format